# Novel Mg-Incorporated Micro-Arc Oxidation Coatings for Orthopedic Implants Application

**DOI:** 10.3390/ma14195710

**Published:** 2021-09-30

**Authors:** Rongfa Zhang, Sheng Zhong, Lilan Zeng, Hongyu Li, Rongfang Zhao, Shufang Zhang, Xinting Duan, Jingsong Huang, Ying Zhao

**Affiliations:** 1School of Materials and Electromechanics, Jiangxi Science and Technology Normal University, Nanchang 330038, China; zs18200478919@163.com (S.Z.); zhaorfamy@126.com (R.Z.); zhang63793@163.com (S.Z.); dxt1026@163.com (X.D.); h18370520439@163.com (J.H.); 2Shenzhen Institute of Advanced Technology, Chinese Academy of Sciences, Shenzhen 518055, China; xiaomiao7525@163.com (L.Z.); hy.li1@siat.ac.cn (H.L.)

**Keywords:** titanium alloys, micro-arc oxidation, cytocompatibility, magnesium, H_12_Phy

## Abstract

In this study, Ti-6Al-4V alloy samples were processed by micro-arc oxidation (MAO) in phytic acid (H_12_Phy) electrolytes with the addition of different concentrations of EDTA-MgNa_2_ (Na_2_MgY) and potassium hydroxide (KOH). The surface characterization and cytocompatibility of MAO-treated samples were evaluated systematically. H_12_Phy is a necessary agent for MAO coating formation, and the addition of Na_2_MgY and KOH into the electrolytes increases the surface roughness, micropore size and Mg contents in the coatings. The MAO coatings are primarily composed of anatase, rutile, MgO and Mg_3_(PO_4_)_2_. Magnesium (Mg) ions in the electrolytes enter into MAO coatings by diffusion and electromigration. The MAO coatings containing 2.97 at% Mg show excellent cell viability, adhesion, proliferation, alkaline phosphatase activity, extracellular matrix (ECM) mineralization and collagen secretion, but the cytocompatibility of the MAO coatings containing 6.82 at% Mg was the worst due to the excessively high Mg content. Our results revealed that MAO coatings with proper Mg contents improve the cytocompatibility of the Ti-6Al-4V alloys and have large potential in orthopedic applications.

## 1. Introduction

Titanium (Ti) and its alloys are widely used as metallic implant materials due to high strength, low density and excellent corrosion resistance [1,2,3], but the materials cannot provide sufficient osseointegration with the surrounding bones in vivo. Hence, implant loosening and failure may occur, and it is sometimes necessary to perform surface treatment to improve the biological properties [1,3]. Micro-arc oxidation (MAO), a simple and effective electrochemical technique, can produce porous, relatively rough and adherent anodic coatings on Ti alloys [4,5,6]. The chemical composition of the MAO coatings depends on the electrolyte constituents and processing conditions [4,7,8,9,10,11]. It has been shown that Ti implants with MAO coatings containing a phosphorus (P) element can improve cell adhesion and proliferation [11], and P-containing electrolytes such as H_3_PO_4_ [8,10,11], NaH_2_PO_4_ [3,12], Na_3_PO_4_ [9], sodium hexametaphosphate [13,14], calcium glycerophosphate [15], β-glycerophosphate disodium [16] and phytic acid (C_6_H_6_(PO_4_)_6_H_12_, abbreviated as H_12_Phy or InsP6) [17] have been used in MAO treatment. H_12_Phy, also known as inositol hexakisphosphate, is a natural and nontoxic organic macromolecule, and its structure is shown in Figure 1a [18]. Owing to the structure consisting of active oxygen ligand atoms, H_12_Phy has a strong chelating capability with Ca^2+^ and Mg^2+^, especially in alkaline solutions, to form stable metal-H_12_Phy complexes [18,19,20]. Compared to Na_2_EDTA, a widely used chelating agent, H_12_Phy is less cytotoxic and does not affect the viability and ALP activity of MC3T3-E1 cells [20]. Hence, it is a suitable anticancer agent, a food antioxidant and also functions as an inhibitor for renal stone development [18,21,22].

Besides the P element, some trace elements in natural bone are helpful for bone formation. As the second-most abundant intracellular cation [23], magnesium (Mg) influences many cellular functions, including the transport of potassium and calcium ions, as well as the modulation of signal transduction, energy metabolism and cell proliferation [24,25,26,27,28]. Mg has been incorporated into biomedical implants by MAO treatment [17,28,29,30]. In vivo experiments show that, compared to machine-turned titanium implants or oxidized TiO implants, Mg-incorporated implants (MgTiO) exhibit better bone integration [28]. In addition, the immersion tests in simulated body fluid (SBF) indicate that the presence of Mg in oxide layers enhances the coating bioactivity [29]. However, the mechanism of Mg incorporation into MAO coatings is not clear. In addition, the correlation between the Mg amount in MAO coatings and its in vitro cytocompatibility is rarely reported.

In this work, in order to investigate the influences of the electrolyte constituents on the Mg contents and surface properties of anodic coatings, MAO coatings were prepared on Ti-6A1-4V alloys with electrolytes containing 15-g/L H_12_Phyand different concentrationsofethylene diaminetetraacetic acid magnesium disodium salt (C_10_H_12_N_2_O_8_Na_2_Mg, EDTA-MgNa_2_, abbreviated as Na_2_MgY) and potassium hydroxide (KOH). The underlying mechanism of the Mg ions incorporation into MAO coatings on titanium alloys was firstly investigated. In addition, the surface characterization and in vitro cytocompatibility of the MAO-treated samples with different Mg contents were evaluated systematically.

## 2. Materials and Methods

### 2.1. Micro-Arc Oxidation Treatment

Widely used medical Ti-6Al-4V alloy samples without annealing or deformation were machined into a cuboidal shape (10 mm× 10 mm × 2 mm or 50 mm × 50 mm × 2 mm). The samples were progressively ground with sandpaper from 80 to 1000 grit, washed with distilled water, dried under hot air and stored in a desiccator prior to MAO treatment. The electrolytes were composed of 15-g/L H_12_Phyand 5, 10, 15 and 20-g/L Na_2_MgY and designated as the S-Mg-5 system, S-Mg-10 system, S-Mg-15 system and S-Mg-20 system, respectively. The corresponding MAO samples fabricated in the four solutions outlined above were denoted as the Mg-5 system, Mg-10 system, Mg-15 system and Mg-20 system, respectively. In order to study the influences of the solution conductivity on the coating properties, 2, 5, 8 and 11-g/L KOH were separately added into the S-Mg-10 system, a solution with moderate Na_2_MgY concentration, and these solutions were labeled as S-the Mg-10-KOH-2 system, S-Mg-10-KOH-5 system, S-Mg-10-KOH-8 system and S-Mg-10-KOH-11 system, respectively. The aqueous electrolyte solutions and the corresponding MAO coatings are listed in Table 1.

The solution conductivity was measured by a DDS-307W microprocessor conductivity meter (Shanghai LIDA Instrument Factory, Shanghai, China). A homemade MAO-50C powder supply under the constant current control mode was used for MAO treatment. A Ti-6Al-4V sample and a stainless-steel barrel containing the electrolyte were separately served as the anode and the cathode. The schematic diagram of MAO treatment is shown in Figure 1b. MAO was performed at a current density of 50mA/cm^2^, duty cycle of 35% and pulse frequency of 2000 Hz for 3 min.

### 2.2. Surface Characterization

The surface morphology and chemical composition of the samples were examined on ascanning electron microscope (SEM, Zeiss ΣIGMA, Oberkochen, Germany) with an energy-dispersive X-ray spectroscopy (EDS, OxfordINCA Energy, Oxford, UK) attachment by a secondary electron detector using an accelerating voltage of 20 kV. The constituent-phase structure of the samples was determined by X-ray diffraction (XRD, Shimadzu XRD-6100, Kyoto, Japan) with Cu K_α_ in a scanning range between 10° and 80°. An X-ray photoelectron spectroscopy (XPS, ESCALAB250, Thermo VG, Waltham, NV, USA) with an Al K_α_ (λ = 1486.6 eV) was used to determine the chemical states of thesamples after sputtering for 60 s to remove the surface contaminants. The binding energies were referenced to the C 1s line at 284.6 eV.

### 2.3. The Measurement of Mg^2+^ Concentration

According to our previous experiments, the increasing KOH concentration could significantly improve the Mg content. In order to clarify the underlying mechanism of Mg ion incorporation into MAO coatings, in one solution with a high KOH concentration—for example, the S-Mg-10-KOH-8 system—two syringes around the anode and around the cathode were used to simultaneously fetch the solution during MAO treatment for 30 s and 150 s (Figure 1b). The Mg ion concentration was analyzed by an inductively coupled plasma optical emission spectrometry (ICP-OES) (PE Optima8300, Perkin-Elmer Corporation, Waltham, NV, USA) with the analyzed wavelength of 285.213 nm.

### 2.4. Cell Culture and Viability

The mouse pre-osteoblast cells (MC3T3-E1, ATCC CRL-2594) were cultured in alpha-minimum essential medium (α-MEM, Gibco, OK, USA) supplemented with 10% fetal bovine serum (FBS, Gibco, OK, USA) and 100 units/mL of penicillin and streptomycin at 37 °C in a 5% CO_2_ humidified atmosphere. Prior to the experiments, all the samples were sterilized by 70% ethanol for 30 min and rinsed with sterile phosphate-buffered saline (PBS, HyClone, Marlborough, MA, USA) three times. The cytotoxicity of the samples was evaluated using the Cell Counting Kit-8 (CCK-8, Beyotime, Haimen, China). The extract was prepared based on a sample surface area-to-extraction medium volume ratio of 1 cm^2^/mL for 3 days. The MC3T3-E1 cellswith an initial density of 5.0 × 10^3^ cells/well were seeded on 96-well plates. After culturing for 24 h, 100-μLextract supplemented with 10% FBS were added into each well to replace the initialα-MEM. After further incubation for 1, 3 and 7days, the α-MEM containing 10% CCK-8 was added into each well and incubated at 37 °C for another 2 h. Afterwards, the optical density was monitored at 450 nm on a microplate reader (Thermo Fisher Scientific, Waltham, NV, USA), and the cell viability (%) was determined according to the manufacturer’s instructions.

### 2.5. Initial Cell Attachment and Cell Proliferation

To assess the initial cell attachment and proliferation, the MC3T3-E1 cells were seeded directly on each sample with a density of 1 × 10^4^ cells/well on 24-well plates. During the initial cell attachment, the cells were allowed to settle for 4 h. The seeded samples were then washed twice with PBS and fixed for 5 min with 3.7% formaldehyde solution in PBS. The cytoskeleton protein F-actin was stained with phalloidin-fluorescein isothiocyanate (Invitrogen^TM^, Ronkonkoma, NY, USA) for 40 min at room temperature, and the nuclei were counterstained with DAPI (Sigma, Ronkonkoma, NY, USA). The cell images were captured by a fluorescence microscope (Olympus-IX71, Shinjuku City, Japan). Cell proliferation was evaluated using CCK-8 (Beyotime, China) after the cells were cultured on the Ti-6A1-4V samples for 1, 3 and 7 days. At each timepoint, the samples were rinsed with PBS and transferred to new 24-well plates. The α-MEM containing 10% CCK-8 was added into each well and incubated at 37 °C for another 2 h. Then, 100 μL of solution was transferred to new 96-well plates. The absorbance was measured at a wavelength of 450 nm on a microplate spectrophotometer (Thermo Fisher Scientific, Waltham, NV, USA).

### 2.6. Alkaline Phosphatase (ALP) Activity

ALP activity was evaluated with an Alkaline Phosphatase Assay kit (Beyotime, China) to determine the pre-osteoblast differentiation properties. The MC3T3-E1 cells were seeded on the Ti-6A1-4V samples with a density of 1 × 10^4^ cells/well and incubated for 1 day. On the second day, α-MEM supplemented with 50-μg/mL ascorbic acid (Sigma, Ronkonkoma, NY, USA) was added into each well to replace the initialα-MEM, and after culturing for 7 days, 10-mM b-glycerol phosphate(Sigma, Ronkonkoma, NY, USA) was added together with ascorbic acid. After 3, 7 and 14 days, the seeded cells were washed twice with PBS and lysed in the M-PER Reagent (Thermo Scientific, Waltham, NV, USA) at 4 °C for 10 min. The cell lysates were then centrifuged at 14,000× *g* for 10 min. Fifty microliters of supernatant were transferred to a new 96-well culture plate and mixed with 50 μL of ALP reagent containing p-nitrophenyl phosphate (p-NPP) as the substrate. Finally, the absorbance was recorded on a spectrophotometer at 405 nm. The total proteins in the cell lysate were determined by a bicinchoninic acid (BCA) protein assay kit (Pierce, Rockford, Waltham, NV, USA). The ALP activity was normalized to the total protein level of the samples, and the data were expressed as the specific ALP activity per unit of proteins.

### 2.7. Extracellular Matrix (ECM) Mineralization and Collagen Secretion

Extracellular matrix mineralization (ECM) on and collagen secretion by the MC3T3-E1 cells on the Ti-6A1-4V samples were assessed using Alizarin Red and Sirius Red staining, respectively. After culturing for 21 days, the cells with an initial density of 1 × 10^4^/well were washed and fixed. The cells were incubated with 1-mg/mL Alizarin Red S (Sigma-Aldrich, Ronkonkoma, NY, USA) or 0.1% Sirius Red solution (Sigma, Ronkonkoma, NY, USA) in saturated picric acid for 30min to reveal ECM mineralization or collagen secretion, respectively. After rinsing with deionized water, the stained cells were taken by a digital camera (Nikon D3200, Tokyo, Japan). Afterwards, the Alizarin Red or Sirius Red stain was dissolved in 10% cetylpyridinum chloride in 10-mM sodium phosphate (pH 7) or 0.2-M NaOH/methanol (1:1), and the absorbance was measured at a wavelength of 620 nm or 540 nm on a microplate reader (Thermo Scientific Appliskan, Waltham, NV, USA).

### 2.8. Statistical Analysis

Each experiment was repeated three times, and the data were presented as the mean ± SD. The differences were compared using the unpaired Student’s *t*-test, and a value of *p* < 0.05 was considered to be statistically significant.

## 3. Results

### 3.1. Influences of Na_2_MgY Concentrationon on Solution Conductivity and MAO Coatings

The conductivity of the solution containing 15-g/L H_12_Phy and 5-g/L Na_2_MgY (S-Mg-5 system) at 10 °C was 4.02 mS/cm. When the Na_2_MgY concentrations were increased to 10, 15 and 20 g/L, the corresponding conductivities changed to 4.55, 5.66 and 6.65 mS/cm, respectively. As shown in Table 2, as the Na_2_MgY concentration went up, the Mg content in the MAO coatings increased. At the same time, the P content initially increased rapidly but then decreased slowly, and the Mg-15 system achieved the maximum P content.

The surface morphology and phase structure of the samples prepared in solutions with different Na_2_MgYconcentrations are presented in Figure 2a–d and Figure 3a. The anodized Ti-6A1-4V alloy exhibited a typical porous structure. The pore size of the Mg-5 system and Mg-10 system ranged from 0.1 to 2.0 μm (Figure 2a,b), whereas that of the Mg-15 system varied between 1.0 and 3.0 μm (Figure 2c). The Mg-20 system exhibited uneven micropore characteristics, and the largest pore could achieve 8 μm in size (Figure 2d).

The samples fabricated in the solutions containing 15-g/L H_12_Phy and different Na_2_MgY concentrations were mainly composed of anatase, rutile, MgO and Mg_3_(PO_4_)_2_ phases (Figure 3a). With increasing Na_2_MgY concentrations from 5 to 20 g/L, the characteristic peak of the Mg_3_(PO_4_)_2_ phase at (100) increased in intensity and exhibited the maximum intensity for the Mg-20 system.

### 3.2. Influence of KOH Concentration on Solution Conductivity and MAO Coatings

In order to improve the Mg content in the coatings, KOH was added into the S-Mg-10 system to adjust its solution property. The conductivity of the S-Mg-10 system was 4.55 mS/cm, and after the addition of 2-g/L KOH, the conductivity of the S-Mg-10-KOH-2 system increased to 5.80 mS/cm. Similarly, the conductivity of the S-Mg-10-KOH-5 system, S-Mg-10-KOH-8 system and S-Mg-10-KOH-11 system increased to 8.97, 12.69 and 20.05 mS/cm, respectively. According to Table 2, the Mg contents increased with the KOH concentrations, but the P contents decreased slowly. The Mg-10-KOH-11 system showed the maximum Mg content of 7.13 at%. Compared to the Mg-10 system (Figure 2b), the Mg-10-KOH-2 system achieved a larger pore size and lower pore density (Figure 2e). As the KOH concentration was increased, the fabricated MAO coatings became rougher, with uneven micropore distribution (Figure 2f–h).

As shown in Figure 3b, the Mg-10 system was composed of primarily anatase together with small amounts of rutile, MgO and Mg_3_(PO_4_)_2_. Compared to the Mg-10 system, the addition of KOH increased the intensity of the Mg_3_(PO_4_)_2_ peak, and as the KOH concentration was increased, anatase, rutile and MgO increased gradually. However, with the increasing KOH concentration from 2 to 11 g/L, the ratios of the Mg_3_(PO_4_)_2_ peak intensity at (100) in the MAO coatings to the titanium peak intensity at (002) exhibiteda slowly decreasing trend.

### 3.3. Influences of H_12_Phy on Coating Formation

According to the experiments, the influences of solution conductivity on the coating properties were investigated in the base solution with 15-g/L H_12_Phy and 10-g/L Na_2_MgY (the S-Mg-10 system). In addition, the increasing KOH concentrations could significantly improve the Mg amount. In order to clarify the effect of H_12_Phy on coating formation, the Ti-6Al-4V alloy was anodized for 3 min in a solution containing 10-g/L Na_2_MgYand a high KOH concentration, for example, 8-g/L KOH. The peak voltage could not be higher than 130 V, and the surface morphology and EDS spectra of the anodic coating was shown in Figure 4a,b. It was clear that MAO coatings were not successfully developed on the titanium alloy withoutH_12_Phy, indicating that H_12_Phy was anecessary agent for coating formation.

### 3.4. XPS Analysis

Since theMg-10-KOH-8 system presented a high Mg content, its XPS survey and high-resolution spectra were acquired to clarify the reaction products in the oxide layer, and the results are shown in Figure 5. According to Figure 5a, Mg, Ti, O, C and P were detected. The C 1s spectrum was composed of two subpeaks at 284.6 and 285.7 eV (Figure 5b), with the former attributed to adventitious carbon species or C-C or C-H from H_12_Phy radicals and the latter corresponding to C-O bonds from adsorbed H_12_Phy on the surface or the reaction products phytates [22,31]. The O 1s peaks with binding energies of 530.3 eV, 531.5 eV and 532.7 eV (Figure 5c) were O^2−^, PO_4_^3−^ and C-O or HPO_4_^2−^, respectively [11,31]. Figure 5d showed the Ti 2p doublet spectra of Ti 2p3/2 and Ti 2p1/2, indicating the presence of Ti_2_O_3_ and TiO_2_ at 457.7 eV and 458.7 eV [32,33]. The P 2p peaks (Figure 5e) at 132.8 eV and 133.7 eV revealed the presence PO_4_^3−^and HPO_4_^2−^, respectively [34], and the Mg 1s peaks (Figure 5f) at 1303.8 eV and 1304.5 eV corresponded to MgO [35] and Mg_3_(PO_4_)_2_, respectively.

### 3.5. Mg Ions Concentration Analysis

The Mg ions concentrations in the Mg-10-KOH-8 system before and after MAO treatment are shown in Figure 6. Prior to MAO treatment, the Mg concentration was 681.17± 5.07 mg/L. After anodizing for 30 s and 150 s, the Mg concentrations near the anode were separately 675.63 ± 4.29 mg/L and 667.11 ± 10.01 mg/L, slightly higher than those close to the cathode at 662.89 ± 8.37 mg/L and 660.79 ± 5.12 mg/L, respectively.

### 3.6. Initial Cell Attachment

As shown in Figure 7, the number of MC3T3-E1 cells on the MAO-treated Ti-6A1-4V in the Mg-10-KOH-5 system was greater than those on the untreated control, the Mg-10 system and the Mg-10-KOH-8 system after incubation for 4 h. The cells on all the samples had a polygonal or spindle shape, and the filose pseudopodium and flat membrane were evident. Compared to the other groups, the cells on the Mg-10-KOH-5systemexhibited more spreading and superior filopodia extension, and so, this system could promote initial cell attachment.

### 3.7. Cell Viability

The cell viability assessed by the CCK-8 assay is shown in Figure 8a. All the extracts were well-tolerated by the pre-osteoblasts, with the cell viability ranging from 130% to 95% over time in comparison with the untreated control group ranging from 120% to 100%. At day 1, the viability of the Mg-10 system and Mg-10-KOH-8 system was lower than that of the untreated control group (*p* < 0.05, Figure 8a). At day 3, the viability of the Mg-10 system, Mg-10-KOH-5 system and Mg-10-KOH-8 system was lower than that of the untreated control group (*p* < 0.05, 0.05 and 0.01, respectively). At day 7, the viability of the Mg-10-KOH-8 system was lower than that of the untreated control group (*p* < 0.01, Figure 8a) but still reached over 95%. In comparison, at day 7, the viability of the Mg-10 system and Mg-10-KOH-5 system was similar to that of the untreated control group (*p* > 0.05, Figure 8a).

### 3.8. Cell Proliferation

Gradually increasing the cell proliferations was observed throughout the culturing period from 1, 3 to 7 days (Figure 8b). In particular, the cells on the Mg-10 system and Mg-10-KOH-8 system showed lower proliferation rates than the untreated control at day1 (*p* < 0.05; Figure 8b). At day 3, the proliferation rates of the Mg-10 system, Mg-10-KOH-5 system and Mg-10-KOH-8 system were lower than that of the untreated control group (*p* < 0.05, 0.05 and 0.01, respectively). At day 7, the proliferation rates of the Mg-10-KOH-8 system were lower than that of the untreated control group (*p* < 0.01, Figure 8b), but the proliferation rates of the Mg-10 system and Mg-10-KOH-5 system were similar to that of the untreated control group (*p* > 0.05, Figure 8b).

### 3.9. Cell Differentiation

As shown in Figure 8c, the ALP activities of the MC3T3-E1 cells in all the groups at day 3 were not significantly different but increased afterwards with the culturing time. At days 7 and 14, the ALP activities of the MCT3-E1 cells of the Mg-10-KOH-5 system were higher than that of the untreated control group (*p* < 0.05 for day 7 and *p* < 0.01 for day 14, respectively). The ALP activities of the Mg-10 system and Mg-10-KOH-8 system were similar to that of the untreated control group (*p* > 0.05). ECM mineralization of the MC3T3-E1 cells after culturing for 21 days is shown in Figure 8d,f. As shown in Figure 8d, the mineralized calcium nodules were present on all the samples. Compared to the untreated control, the Mg-10 system, Mg-10-KOH-5 system and Mg-10-KOH-8 system showed more mineralized calcium nodules, suggesting that MAO could promote ECM mineralization. In particular, the Mg-10-KOH-5 system exhibited significantly upregulated ECM mineralization compared to the Mg-10 system and Mg-10-KOH-8 system (Figure 8f). Figure 8e,f presented the collagen secretion after incubation for 21 days. As shown in Figure 8e, the Mg-10 system and Mg-10-KOH-8 system showed slightly more collagen secretion than the untreated control, whereas the Mg-10-KOH-5 system exhibited denser collagen secretion. Figure 8f further confirmed significantly upregulated the collagen secretion for the Mg-10-KOH-5 system, and the results were consistent with the ECM mineralization results.

## 4. Discussion

In our work, H_12_Phy, a medium-strong acid, was the major ingredient of the MAO electrolytes. H_12_Phy and other inositol phosphates such as InsP4 and InsP5 are ubiquitously involved in cellular signal transduction and regulation in eukaryotic cells [36]. In addition, H_12_Phy is stable in water at a low temperature and used as an environmentally friendly chemical conversion agent for magnesium alloys [18,23,37]. On account of the strong chelating ability of H_12_Phy with Mg ions, magnesium phytate may be developed and absorbed on MAO coatings, which could be inferred according to the C-O bond at 285.7 eV (Figure 5b).

A higher electrolyte concentration produces larger micropores onanodic coatings with rougher surfaces due to the change in the solution conductivity [38,39]. According to the empirical equation proposed by Burger and Wu [40], the breakdown voltage increased almost linearly with the logarithmic electrolyte resistivity. Larger Na_2_MgY and KOH concentrations increased the conductivity of the solutions, resulting in a lower breakdown voltage and more spark discharge [39,40]. Therefore, larger Na_2_MgY or KOH concentrations enlarge the pore size in anodic coatings and yield a rougher surface morphology. Our study also indicated that Na_2_MgY and KOH influence the surface morphology of MAO coatings in a Na_2_MgY and KOH concentration-dependent manner.

Na_2_MgY and KOH also affect the chemical composition of anodic coatings. The Na_2_MgY concentration imposes a larger effect on the Mg and P contents, while KOH shows a larger effect on the Mg content. This influence can be explained according to the MAO characteristics. Similar to magnesium alloys, there are four stages in MAO of titanium alloys—namely, before anodizing, traditional anodizing, MAO treatment and arc anodizing, as shown in the schematic diagram in Figure 9.

In the first stage, H_12_Phy molecules in the aqueous solution are converted into different H_12_Phy radicals, which compete with Y^−^ ions to combine with Mg^2+^ ions, and the following reactions occur [17]:Na_2_MgY⇆MgY^2−^ + 2Na^+^(1)
MgY^2−^⇆Mg^2+^ + Y^4−^(2)
C_6_H_6_(PO_4_)_6_H_12_ + iMg^2+^ + (12-j) OH^−^ = [Mg_i_H_j_C_6_H_6_(PO_4_)_6_]^(12−2i−j)−^ + (12-j)H_2_O(3)

As shown in Figure 1a, one H_12_Phy molecule has 12 hydrolysable hydrogens. In Equation (3), j is the number of left hydrolysable hydrogens (j = 0–12), while i is the number of Mg ions combined with one H_12_Phy molecule (j = 0–6). According to Equations (1) and (2), a small amount of Mg^2+^ is present in the solution. Owing to the strong chelating ability of H_12_Phyradicals with cations such as Mg^2+^, insoluble magnesium phytate may be formed according to Equation (3), and the solution becomes a turbid colloid.

In the second stage, anions in the solution such as OH^−^, Phy^12−^, [Mg_i_H_j_C_6_H_6_(PO_4_)_6_]^(12−2i−j)−^ and MgY^2−^ are driven to the anode by the electric field. In water solutions, cations move to the anode mainly by diffusion [17,22]. In our study, Mg ions were combined with the chelating agents and became negatively charged. Therefore, electromigration drives negatively charged Mg ions to the anode, which was verified by the higher Mg ion concentrations near the anode than that near the cathode (Figure 6). In addition, microsparks are not developed on the anode surface, and so, the temperature on the sample surface is relatively low, and the H_12_Phy radicals are stable. In this stage, a small amount of TiO_2_, Mg(OH)_2_ or MgO may be produced on the anode surface (Equations (4)–(7)):Ti − 4e^−^ = Ti^4+^(4)
Ti^4+^ + 4OH^−^ = TiO_2_ + H_2_O(5)
Mg^2+^ + 2OH^−^ = Mg(OH)_2_(6)
Mg(OH)_2_ = MgO + H_2_O(7)

As time elapses, small and dense sparks can be observed on the anode surface. In the third stage, the instantaneous temperature on the anode surface is estimated to reach 2116–2643 K due to the spark discharge [41]. It is reported that H_12_Phy can be hydrolyzed into small molecular inositol phosphates in the temperature range between 320 and 345 °C [18]. During MAO, H_12_Phy radicals or magnesium phytate at the anode/electrolyte interface are not stable and are hydrolyzed into inorganic phosphates according to Equations (8) and (9): [Mg_i_H_j_C_6_H_6_(PO_4_)_6_]^(12−2i−j)^^−^→Mg_3_(PO_4_)_2_ + PO_4_^3−^ + H_2_O + C_6_H_6_(OH)_6_(8)
3Mg^2+^ + 2PO_4_^3−^ = Mg_3_(PO_4_)_2_(9)

In the fourth stage, there are larger sparks on the sample surface, and the main reactions on the anode surface are similar to those in the third stage.

According to Equations (4)–(9), anatase, rutile, MgO and Mg_3_(PO_4_)_2_ are formed in MAO coatings. With increasing the Na_2_MgY concentration, more Mg^2+^ is present in the solution, according to Equations (1) and (2). The Mg ions take part in the coating formation, according to Equations (8) and (9), and the MAO coatings contain more Mg and P elements, showing that Mg ions enter into anodic coatings by diffusion. Furthermore, with the increasing KOH concentrations, the Mg count increases but that of P decreases. During MAO, OH^−^ competes with H_12_Phy radicals to combine with cations such as Mg^2+^ to form stable compounds. In addition, OH^−^ ions are smaller and move faster thanH_12_Phy radicals, and therefore, more OH^−^ions are present on the anode. Magnesium ions prefer to combine with OH^−^ ions, and thus, more MgO is formed, according to Equations (6) and (7). Therefore, with the increasing KOH concentrations, MAO coatings contain more Mg but less P, exhibiting further increases in the surface roughness.

The cell behaviors, such as attachment, proliferation and differentiation, are mainly determined by the properties of the anodic coatings, such as the composition, roughness, hydrophilicity, morphology and microstructure, which, in turn, depend on the processing parameters, such as the electrolyte composition and concentration, treatment time and final voltage [3,4,10,11,12,38,42,43]. H_12_Phy, found in eukaryotes and an abundant inositol phosphate in cells, can increase the lipid storage capacity, improve the glucose uptake and inhibit lipolysis [44]. In vitro cell tests show that magnesium alloys treated in a H_12_Phy-containing solution show a high cell viability and proliferation [33,37]. Our results demonstrated that the MAO coatings had no cytotoxicity, but the different solutions did affect the cell attachment, proliferation and differentiation. With increasing the Na_2_MgY and KOH concentrations, both the surface roughness and micropore size increased. The Mg-10-KOH-5 system (containing 2.97 at% Mg) showed better cell viability, proliferation and differentiation than the Mg-10 and Mg-10-KOH-8 systems, suggesting that the chemical composition and microstructure of the MAO coating played an important role in the pre-osteoblast performance, in addition to the surface morphology.

Mg is a vital element, and the appropriate Mg concentration induces osteogenic activity [26]. There is Mg_3_(PO_4_)_2_ and MgO in the MAO coatings, and it is well-known that magnesium salts, especially magnesium phosphate, have good biocompatibility [45,46]. With increasing the KOH concentrations, the Mg contents increase while the P concentration decreases slowly, thus resulting in more MgO, producing some deleterious effects [25,47]. Therefore, our data show that the Mg-10-KOH-5 system has better in vitro cytocompatibility than the Mg-10-KOH-8 system.

## 5. Conclusions

Ti-6A1-4V alloy samples were subjected to MAO treatment in an environmentally friendlyorganic P-containing and a novel Mg-containing electrolyte and showed good in vitro cytocompatibility based on the viability, adhesion, proliferation and differentiation of the MC3T3-EI pre-osteoblast. Higher Na_2_MgY and KOH concentrations increased the roughness of the MAO coatings, and the best cytocompatibility wasachieved from H_12_Phy with 10-g/L Na_2_MgY; 5-g/L KOH. H_12_Phy is a key agent for MAO coating formation, and Mg ions enter into MAO coatings mainly by electromigration. The results suggested that both the surface morphology and Mg contents affected the in vitro cytocompatibility of the MAO coatings on Ti-6A1-4V.

## Figures and Tables

**Figure 1 materials-14-05710-f001:**
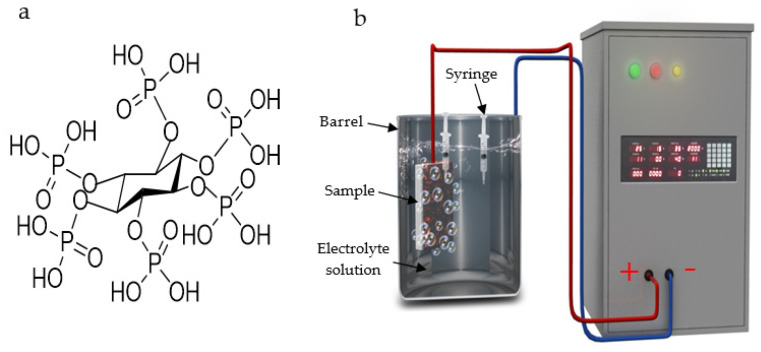
(**a**) H_12_Phy structure. (**b**) The schematic diagram of the MAO experiment.

**Figure 2 materials-14-05710-f002:**
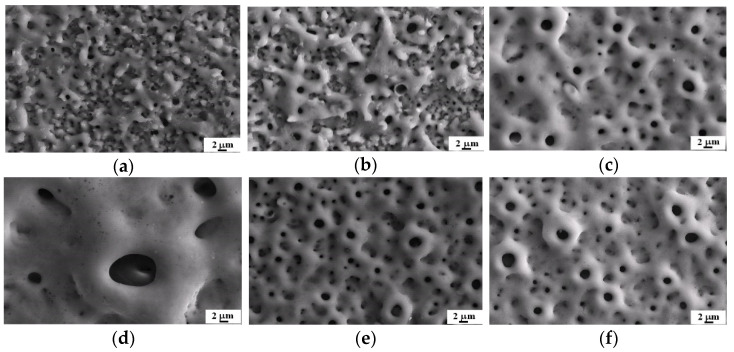
SEM images of the MAO coatings produced in solutions with different Na_2_MgY and KOH concentrations: (**a**) the Mg-5 system, (**b**) Mg-10 system, (**c**) Mg-15 system, (**d**) Mg-20 system, (**e**) Mg-10-KOH-2 system, (**f**) Mg-10-KOH-5 system, (**g**) Mg-10-KOH-8 system and (**h**) Mg-10-KOH-11 system.

**Figure 3 materials-14-05710-f003:**
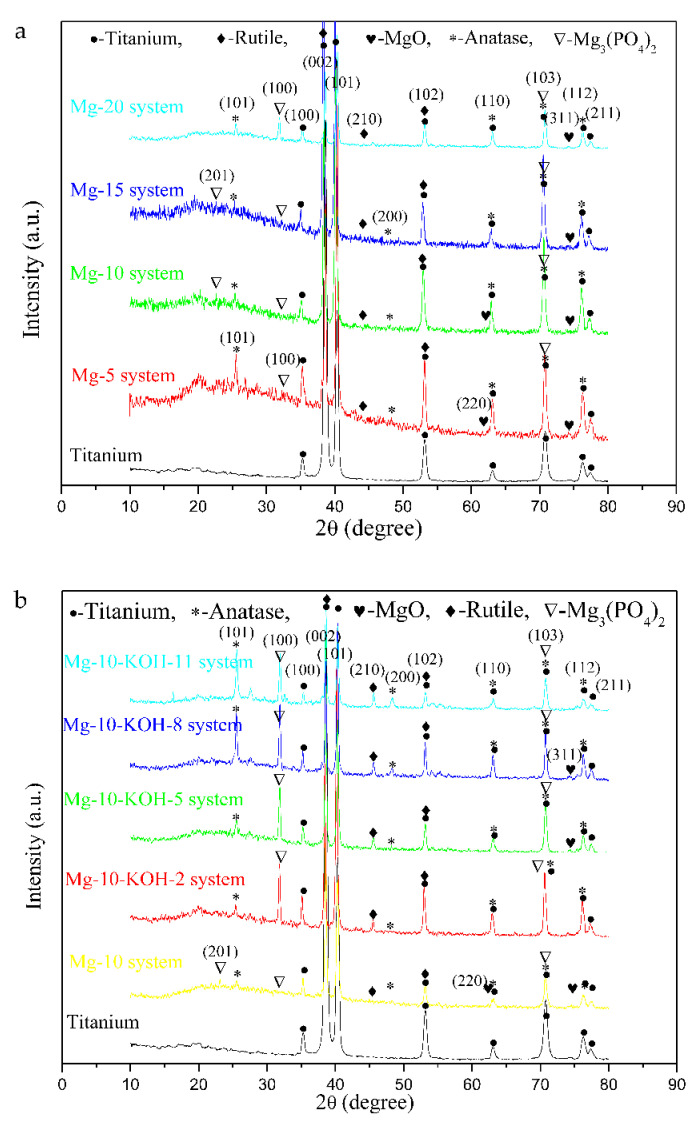
X-ray diffraction patterns of the MAO samples fabricated in (**a**) 15g/L H_12_Phywith 5, 10, 15 and 20 g/L Na_2_MgY and (**b**) 15g/L H_12_Phy and 10 g/L Na_2_MgY without and with 2, 5, 8 and 11 g/L KOH.

**Figure 4 materials-14-05710-f004:**
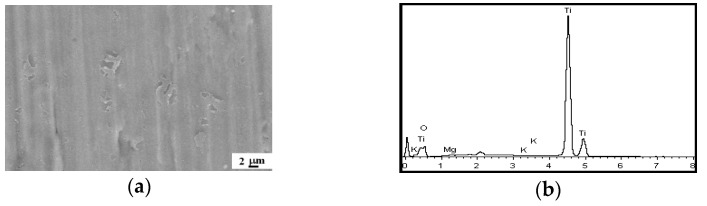
(**a**) The SEM image and (**b**) EDS spectrum of the MAO coatings produced in the solution of 10-g/L Na_2_MgY and 8-g/L KOH.

**Figure 5 materials-14-05710-f005:**
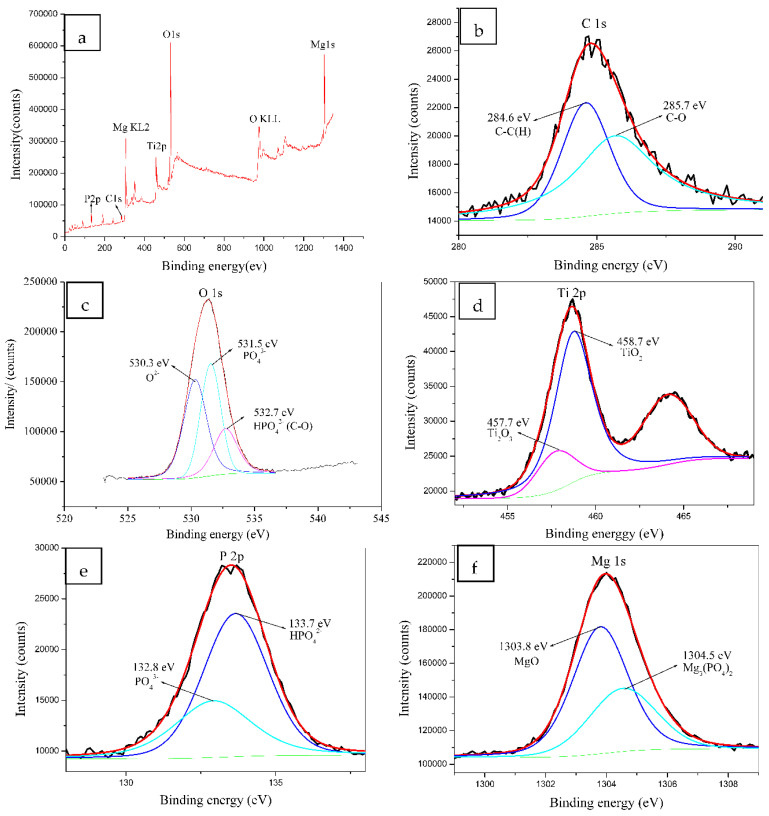
XPS spectra of the anodic coatings obtained at a current density of 50 mA/cm^2^, duty cycle of 35%, pulse frequency of 2000 Hz and treatment time of 3 min in the Mg-10-KOH-8 system: (**a**) survey, (**b**) C 1s, (**c**) O 1s, (**d**) Ti 2p, (**e**) P 2p and (**f**) Mg 1s.

**Figure 6 materials-14-05710-f006:**
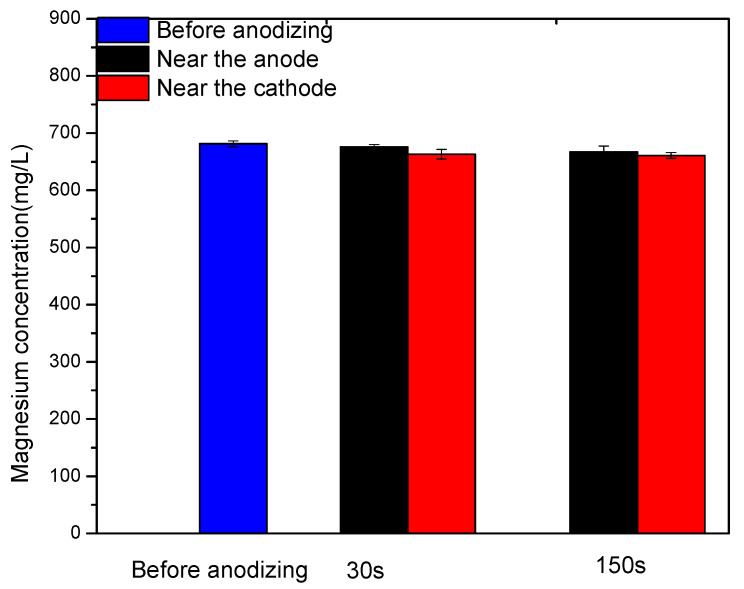
The magnesium concentration in the solution of the Mg-10-KOH-8 system before MAO treatment and near the anode and the cathode after MAO treatment for 30 s and 150 s.

**Figure 7 materials-14-05710-f007:**
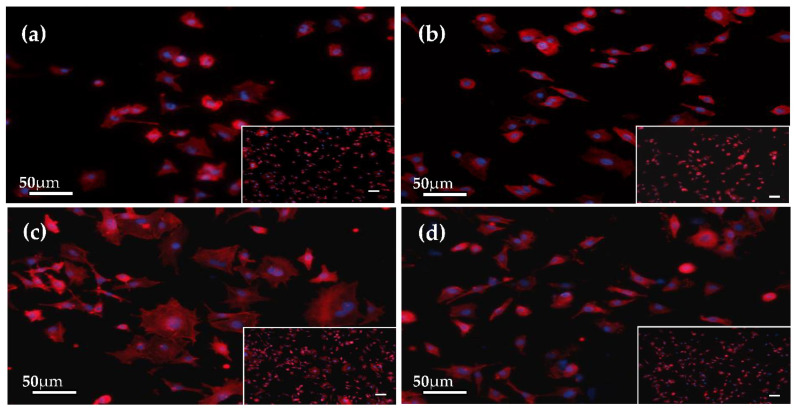
Fluorescent images of the MC3T3-E1 pre-osteoblasts after culturing for 4 h: (**a**) untreated control, (**b**) the Mg-10 system, (**c**) the Mg-10-KOH-5 system and (**d**) the Mg-10-KOH-8 system. The scale bar in the low-magnification insets is 100 μm.

**Figure 8 materials-14-05710-f008:**
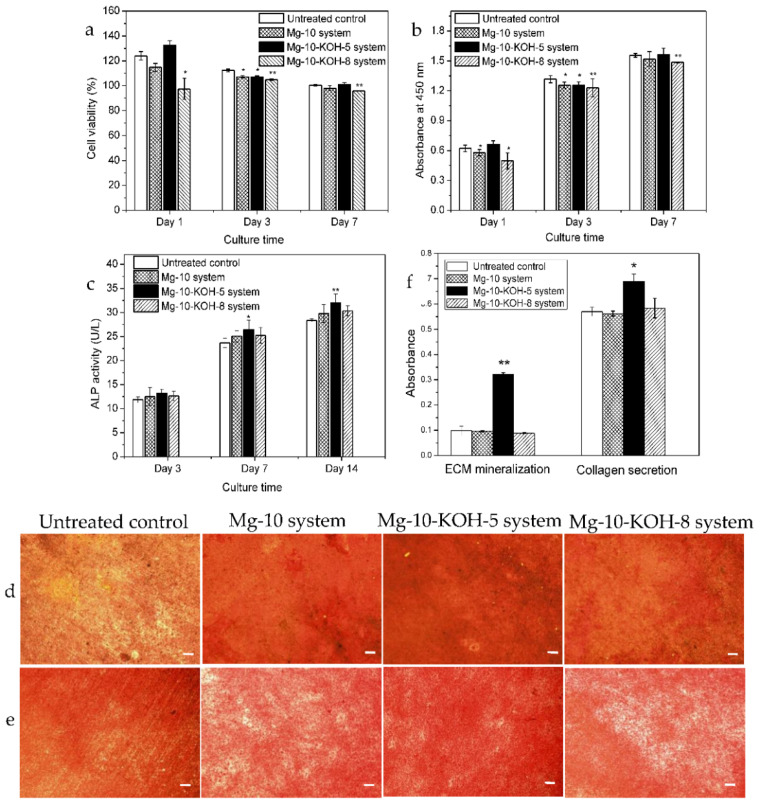
In vitro cytocompatibility evaluation of the MAO samples after culturing MC3T3-E1 pre-osteoblasts for a different period. (**a**) Cell viability, (**b**) cell proliferation, (**c**) ALP activities, (**d**) ECM mineralization, (**e**) collagen secretion and (**f**) the quantitative expression of ECM mineralization and collagen secretion * *p* < 0.05 and ** *p* < 0.01 versus the untreated control. Scale bar is 200 μm.

**Figure 9 materials-14-05710-f009:**
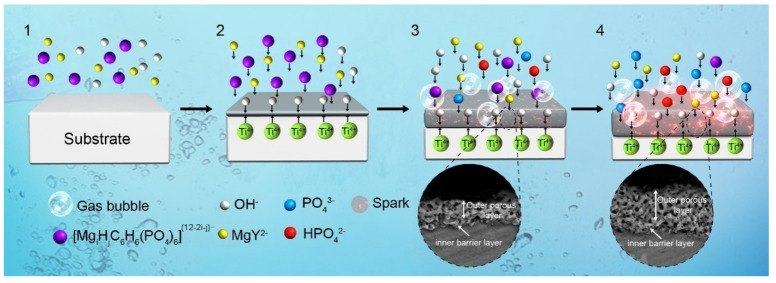
Schematic diagram showing the four stages of anodized coating formation and growth: (1) before anodizing, (2) traditional anodizing, (3) MAO treatment and (4) arc anodizing.

**Table 1 materials-14-05710-t001:** The fabricated MAO coatings and the corresponding electrolytes compositions.

Processes	Coatings	Electrolyte Concentration (g/L)		
		H_12_Phy	Na_2_MgY	KOH
S-Mg-5 system	Mg-5 system	15	5	
S-Mg-10 system	Mg-10 system	15	10	
S-Mg-15 system	Mg-15 system	15	15	
S-Mg-20 system	Mg-20 system	15	20	
S-Mg-10-KOH-2 system	Mg-10-KOH-2 system	15	10	2
S-Mg-10-KOH-5 system	Mg-10-KOH-5 system	15	10	5
S-Mg-10-KOH-8 system	Mg-10-KOH-8 system	15	10	8
S-Mg-10-KOH-11 system	Mg-10-KOH-11 system	15	10	11

**Table 2 materials-14-05710-t002:** Chemical composition of MAO coatings obtained in solutions composed of 15-g/L H_12_Phy and various concentrations (5, 10, 15 and 20 g/L) of Na_2_MgY, and the S-Mg-10 system was added with 2, 5, 8 and 11-g/L KOH.

Processes	Element Content (at%)						
	C	Na	K	O	P	Ti	Mg
Mg-5 system	2.43			60.37	4.80	32.40	
Mg-10 system	3.76			61.76	9.23	24.68	0.57
Mg-15 system	4.94			59.73	10.69	22.80	1.84
Mg-20 system	3.99			60.97	10.51	22.19	2.34
Mg-10-KOH-2 system	5.00			60.97	10.44	21.71	1.87
Mg-10-KOH-5 system	4.17			60.47	9.91	22.50	2.97
Mg-10-KOH-8 system	8.02		0.21	59.03	8.57	17.36	6.82
Mg-10-KOH-11 system	6.34	0.39	0.45	58.28	7.49	19.92	7.13

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
