# Peer review of "Novel Mg-Incorporated Micro-Arc Oxidation Coatings for Orthopedic Implants Application"

_materials, 2021, doi:10.3390/ma14195710_

Round 1

Reviewer 1 Report

The paper presents a very wide range of experimental results regarding Mg-incorporated MAO coatings. 

It needs some improvements in presentation suggested below:

1) in line 56 and 64 it should clearly say what mechanisms is not clear 

2) The XPS analysis and Mg ion concentrations are performed only in Mg10KOH-8 system. Why ?  

3)In figure 2 there is a diagram probably EDS analysis which is not described 

4) Table 1, fig2 and fig 3 are for sections 3.1, 3.2, and 3.3. I suggest to modified that. 

5) similar in sections 3.7, 3.8 and 3.9, description in the fig. 7 is lack of image f), ALP activity is described in 3.9 Cel differentiation paragraph. it needs to be improved. 

Author Response

We highly appreciate the reviewers carefulness, conscientious, and the broad knowledge on the relevant research fields. According to the reviewers advice, we have made the following revisions.

The paper presents a very wide range of experimental results regarding Mg-incorporated MAO coatings.

It needs some improvements in presentation suggested below:

1) in line 56 and 64 it should clearly say what mechanisms is not clear

The part of “the underlying mechanism of Mg ions into MAO coatings” has been changed into “the underlying mechanism of Mg ions incorporation into MAO coatings”.

2) The XPS analysis and Mg ion concentrations are performed only in Mg10KOH-8 system. Why ? 

Because Mg-10-KOH-8 system achieved high Mg content, its XPS survey and high-resolution spectra were acquired to clarify the reaction products in the oxide layer.

3)In figure 2 there is a diagram probably EDS analysis which is not described

Figure 2 has been changed into two figures and EDS analysis in present Figure 4 has been described.

4) Table 1, fig2 and fig 3 are for sections 3.1, 3.2, and 3.3. I suggest to modified that.

Figure 2 has been changed into Figure 2 and Figure 4. In addition, the corresponding contents have been revised.

5) similar in sections 3.7, 3.8 and 3.9, description in the fig. 7 is lack of image f), ALP activity is described in 3.9 Cel differentiation paragraph. it needs to be improved.

The description in the present Figure 8f has been provided as the following: Quantitive expression of ECM mineralization and collagen secretion.

Reviewer 2 Report

This kind of alloy showed good cell adhesion,proliferation and differentiation.The proper Mg concentretaion is is essential for these processes.The style is excellent and in good scientific level.

Author Response

Thank you very much for the comment.

Reviewer 3 Report

This paper deals with the surface modification of titanium alloys by MAO to implant Mg and P and studies the effect of this treatment on biocompatibility. 

The introduction depicts a clear state of the art, experiments are adequately described and executed, and results firmly support the conclusion. I think the paper is excellent and deserves to be published.

I only suggest to check a few points:

  • The authors repeatedly use the phrase "the mechanism of Mg into MAO coatings (lines 56, 64, and 99). I think it is not accurate and further specification (e.g. "the mechanism of incorporation of Mg" could bo better) is necessary;
  • In line 123 there is a typo, "ells" instead of "cells";
  • In lines 182-183, the plural "Figures" is used for a single image;
  • At the bottom of Figure 2 a spectrum is reported, but neither the text nor the caption say anything about it. Please add description for the spectrum or remove it;
  • In line 217 "improved" is used instead of "improve";
  • In line 221 "clearly" is used instead of "clear";
  • In Figure 4, the two bottom spectra (panels e and f) should have the same format as the other ones;
  • In line 243-245, the authors state that "Mg concentrations near the anode were [...] higher than those close to the cathode". However, average values do not seem to differ statistically if we consider the reported errors. Please clarify this aspect;
  • I think the titles of subsections 3.8 and 3.9 are reversed, please check;
  • In line 354, Equation 8 would look better in a single line.

Author Response

We highly appreciate the reviewers carefulness, conscientious, and the broad knowledge on the relevant research fields. According to the reviewers advice, we have made the following revisions.

This paper deals with the surface modification of titanium alloys by MAO to implant Mg and P and studies the effect of this treatment on biocompatibility.

The introduction depicts a clear state of the art, experiments are adequately described and executed, and results firmly support the conclusion. I think the paper is excellent and deserves to be published.

I only suggest to check a few points:

1)The authors repeatedly use the phrase "the mechanism of Mg into MAO coatings (lines 56, 64, and 99). I think it is not accurate and further specification (e.g. "the mechanism of incorporation of Mg" could bo better) is necessary;

The part of “the underlying mechanism of Mg ions into MAO coatings” has been changed into “the underlying mechanism of Mg ions incorporation into MAO coatings”.

2)In line 123 there is a typo, "ells" instead of "cells";

The word “ells” has been changed into “cells”.

3)In lines 182-183, the plural "Figures" is used for a single image;

The sentence of “whereas that of Mg-15 system varied between 1.0 and 3.0mm (Figures 2c)” has been changed into “whereas that of Mg-15 system varied between 1.0 and 3.0mm (Figure 2c)”.

4)At the bottom of Figure 2 a spectrum is reported, but neither the text nor the caption say anything about it. Please add description for the spectrum or remove it;

Figure 2 has been changed into two figures and EDS analysis in present Figure 4 has been described.

5)In line 217 "improved" is used instead of "improve";

The word “improved” has been changed into “improve”.

6)In line 221 "clearly" is used instead of "clear";

The word “clearly” has been changed into “clear”.

7)In Figure 4, the two bottom spectra (panels e and f) should have the same format as the other ones;

The initial XPS results:

Changing into:

8)In line 243-245, the authors state that "Mg concentrations near the anode were [...] higher than those close to the cathode". However, average values do not seem to differ statistically if we consider the reported errors. Please clarify this aspect;

In the experiment, the concentration near the anode is not evidently higher than that close to the cathode. We think that the reasons may be followings. Firstly, the distance between the anode and the cathode is about 10 cm. Secondly, the solution achieves good conductivity and Na2MgY exhibits good solubility. In the future, the similar experiment can be designed by changing the distance between the anode and the cathode. The corresponding contents in the part is changed into “After anodizing for 30 s and 150 s, the Mg concentrations near the anode were separately 675.63±4.29 mg/L and 667.11±10.01 mg/L, slightly higher than those close to the cathode, 662.89±8.37 mg/L and 660.79±5.12 mg/L, respectively”.

9) I think the titles of subsections 3.8 and 3.9 are reversed, please check;

Thank you for the comment. That is a mistake. 3.8 should be revised to cell proliferation and we have revised accordingly.

10)In line 354, Equation 8 would look better in a single line.

Equation has been changed into:

[MgiHjC6H6(PO4)6](12-2i-j)®Mg3(PO4)2 + PO43-+ H2O+C6H6(OH)6

(8)

Reviewer 4 Report

Some minor comments

1. Introduction (line 25-66, p.2)

I should say that introduction part is good. It is clearly reviewed the problem and purpose.

2. Materials and Methods (lines 67-166, p.2 and 4)

Line 69, p.2: the authors could not provide information about the Ti-6Al-4V alloy regarding to alloy compositions, melting and thermomechanical processing. Were the samples used in all experiments received as-annealed or deformed condition?

2.1. MAO treatment

Line 72-80, p.2:  I recommend it should summarize all texts in one table to describe MAO treatment experiments. It will be more easier to understand how the experiments performed.

2.2. Surface characterization

Why the authors could not use Electron BackScattered Diffraction (EBSD technique) to study the surface morphology and chemical composition of the samples?

2.7.  Extracellular...

Line: 158 p.4

were air-dried before.. (Why did not use a vacuume drying under Ar-ion protection) because Ti-Alloy has a high affinity to oxygen (O).

3. Results

3.1. Influence.. 

Figure 2 (a-i): No information provided about SEM such as voltage, detector-type

Between line 188 and 189, p. 5: there is a diagram, but not information about it and not labeled. 

Figure 3 (a and b), P 6:

No data or scale on y-axis

3.2. Influence ...(line 195-208, p.6)

How did pore densities estimated for Figure 2e? could the author say how much percentage % A-system lower than B-system.

3.5. Mg ions.. (line 240, p.8)

could the author say how much percentage % A-system higher than those B-system

Figure 5. (p.9)

Figure 3 shows no significance difference regarding Mg-concentration (mg/L) as a function of before MAO treatment for dwell time 30 and 150 seconds.

3.6. Initial ... (line 249-256, p.9)

...was greater than those.. (How did the number be estimated and could the author say how much percentage % or quantify it instead to say A-system greater than B-system)

3.7. Cell ..(line 261-277, p.10)

could the author say how much percentage % A-system lower than B-system in line 265, 267 and 269

Line 169: was lower than that that of .. (could the author remove one word of that) 

Line 281, 283, and 285 lower mentioned three times and could be added  how much percentage % A-system lower than that B-system

3.9. Cell.. (line 288-305, p.11)

How did the author know A-system higher than B-system (line 291) without given how percentage % .

Line 302: Not clear from Figure 7e. The images are not  visualized by naked eye.

4. Discussion (line 306-390, p.12 and 13)

j and i in Equ. 3 not defined, also what stand j and i for.

Figure 8 shows schematic diagram showing three types of different ball size marked with different colours. Is it right that white ball smaller or similar size compared with a yellow ball.

Line 361 p.13

...coatings by diffusion. Is there any proof for Mg-ions observed inwards Ti-substrate? Why the authors could not use adsorption instead diffusion word.

Author Response

We highly appreciate the reviewers carefulness, conscientious, and the broad knowledge on the relevant research fields. According to the reviewers advice, we have made the following revisions.

Some minor comments

  1. Introduction (line 25-66, p.2)

I should say that introduction part is good. It is clearly reviewed the problem andpurpose.

  1. Materials and Methods (lines 67-166, p.2 and 4)

Line 69, p.2: the authors could not provide information about the Ti-6Al-4V alloy regarding to alloy compositions, melting and thermomechanical processing. Were the samples used in all experiments received as-annealed or deformed condition?

The sentence of “The Ti-6Al-4V alloy samples were machined into a cuboidal shape (10 mm´10 mm´2 mm or 50 mm´ 50 mm´2 mm)” has been changed into “Widely used medical Ti-6Al-4V alloys without annealing or deformation were machined into a cuboidal shape (10 mm´10 mm´2 mm or 50 mm´ 50 mm´2 mm)”.

2.1. MAO treatment

Line 72-80, p.2:  I recommend it should summarize all texts in one table to describe MAO treatment experiments. It will be more easier to understand how the experiments performed.

One table has been added in the revised manuscript.

Thank you for the comments. We accept your advice and Table 1 was added in the revised manuscript.

Table 1 The fabricated MAO coatings and the corresponding electrolytes compositions

Processes

Coatings

Electrolyte concentration (g/L)

H12Phy

Na2MgY

KOH

S-Mg-5 system

Mg-5 system

15

5

S-Mg-10 system

Mg-10 system

15

10

S-Mg-15 system

Mg-15 system

15

15

S-Mg-20 system

Mg-20 system

15

20

S-Mg-10-KOH-2 system

Mg-10-KOH-2 system

15

10

2

S-Mg-10-KOH-5 system

Mg-10-KOH-5 system

15

10

5

S-Mg-10-KOH-8 system

Mg-10-KOH-8 system

15

10

8

S-Mg-10-KOH-11 system

Mg-10-KOH-11 system

15

10

11

2.2. Surface characterization

Why the authors could not use Electron BackScattered Diffraction (EBSD technique) to study the surface morphology and chemical composition of the samples?

In our opinion, secondary electron detector is mainly used to detect surface roughness and cracks; while Electron Back Scattered Diffraction (EBSD technique) is mainly used to detect elemental distribution and cross-sectional morphology. In our study, surface characteristics are mainly detected and therefore EBSD technique is not used.

2.7.  Extracellular...

Line: 158 p.4

were air-dried before.. (Why did not use a vacuume drying under Ar-ion protection) because Ti-Alloy has a high affinity to oxygen (O).

Thank you for the comments. Our expression may confuse you. Actually after rinsing with deionized water, we only waited for several minutes and then took pictures. Since the cells were not fixed on the surface, very dry materials surface would lead to the cells to be dehydrated and the shape change. Original sentence has been changed to “After rinsing with deionized water, the stained cells were taken by a digital camera (Nikon D3200, Japan).

  1. Results

3.1. Influence.. 

Figure 2 (a-i): No information provided about SEM such as voltage, detector-type.

The sentence of “The surface morphology and chemical composition of the samples were examined on a scanning electron microscope (SEM, Zeiss ΣIGMA, Germany) with an energy-dispersive X-ray spectroscopy (EDS) attachment” has been changed into “The surface morphology and chemical composition of the samples were examined on a scanning electron microscope (SEM, Zeiss ΣIGMA, Germany) with an energy-dispersive X-ray spectroscopy (EDS) attachment by secondary electron detector using an accelerating voltage of 20 kV”

Detector type was

Between line 188 and 189, p. 5: there is a diagram, but not information about it and not labeled.

Figure 2 has been changed into two figures and EDS analysis in present Figure 4 has been described.

Figure 3 (a and b), P 6:

No data or scale on y-axis

In the study, XRD was mainly used to detect the phase structure and qualitatively compare the content. Therefore, there was no data or scale on y-axis.

3.2. Influence ...(line 195-208, p.6)

How did pore densities estimated for Figure 2e? could the author say how much percentage % A-system lower than B-system.

In the study, pore densities of MAO coatings are estimated qualitatively and therefore the percentage cannot be achieved. In the future, the coating porosity will be measured using Image-Pro Plus 6.0 imaging software.

3.5. Mg ions.. (line 240, p.8)

could the author say how much percentage % A-system higher than those B-system

After anodizing for 30 s and 150 s, the Mg concentrations near the anode were separately 675.63±4.29 mg/L and 667.11±10.01 mg/L, slightly higher than those close to the cathode, 662.89±8.37 mg/L and 660.79±5.12 mg/L, respectively. We could not provide the exact percentage.

Figure 5. (p.9)

Figure 3 shows no significance difference regarding Mg-concentration (mg/L) as a function of before MAO treatment for dwell time 30 and 150 seconds.

In the experiment, the concentration near the anode is not evidently higher than that close to the cathode. We think that the reasons may be followings. Firstly, the distance between the anode and the cathode is about 10 cm. Secondly, the solution achieves good conductivity and Na2MgY exhibits good solubility. In the future, the similar experiment can be designed by changing the distance between the anode and the cathode. The corresponding contents in the part is changed into “After anodizing for 30 s and 150 s, the Mg concentrations near the anode were separately 675.63±4.29 mg/L and 667.11±10.01 mg/L, slightly higher than those close to the cathode, 662.89±8.37 mg/L and 660.79±5.12 mg/L, respectively”.

3.6. Initial ... (line 249-256, p.9)

...was greater than those.. (How did the number be estimated and could the author say how much percentage % or quantify it instead to say A-system greater than B-system)

Thank you for the comments. We observed cell attachment via fluorescent images. It is actually not a quantitive method to determine the attached cell number on the material surface. Quantitive results of attached cells number can be found in Fig. 7b. Statistically analysis is usually used to compare the difference of cell number between different groups.

3.7. Cell ..(line 261-277, p.10)

could the author say how much percentage % A-system lower than B-system in line 265, 267 and 269

Line 169: was lower than that that of .. (could the author remove one word of that) 

The sentence of “At day 7, the viability of Mg-10-KOH-8 system was lower than that that of the untreated control group (p< 0.01, Figure 8a) but still reached over 95%” has been changed into “At day 7, the viability of Mg-10-KOH-8 system was lower than that of the untreated control group (p< 0.01, Figure 8a) but still reached over 95%”.

Line 281, 283, and 285 lower mentioned three times and could be added  how much percentage % A-system lower than that B-system

Thank you for the comments. For in vitro results, statistical analysis is usually used to compare the expression of different groups. P<0.05 means significant difference between the groups. In our results, we used conventional statistical analysis to determine the biological property of experimental group.

3.9. Cell.. (line 288-305, p.11)

How did the author know A-system higher than B-system (line 291) without given how percentage % .

Thank you for the comments. For in vitro results, statistical analysis is usually used to compare the expression of different groups. P<0.05 means significant difference between the groups. In our results, we used conventional statistical analysis to determine the biological property of experimental group.

Line 302: Not clear from Figure 7e. The images are notvisualized by naked eye.

Thank you for the comments. As shown in Figure 7e, Mg-10 system and Mg-10-KOH-8 system showed slightly more collagen secretion than the untreated control, whereas Mg-10-KOH-5 system exhibited denser collagen secretion. Darker color represents denser collagen secretion. The darkness of the color in the images can be visualized by naked eye.

  1. Discussion (line 306-390, p.12 and 13)

j and i in Equ. 3 not defined, also what stand j and i for.

Two sentences have been added in the revised manuscript: As shown in Figure. 1a, one H12Phy molecule has 12 hydrolyzable hydrogens. In Eq.3, j is the left number of hydrolysable hydrogens (j=0-12), while i is the number of Mg ions combined with one H12Phy molecule (j=0-6)

Figure 8 shows schematic diagram showing three types of different ball size marked with different colours. Is it right that white ball smaller or similar size compared with a yellow ball.

In our opinion, because the OH- is smaller than MgY-, the white ball should be smaller than the yellow ball. However, the characteristics were not exhibited in Figure 8.

Line 361 p.13

...coatings by diffusion. Is there any proof for Mg-ions observed inwards Ti-substrate? Why the authors could not use adsorption instead diffusion word.

During MAO treatment, Mg ions can enter into anodic coatings and therefore the concentration of Mg ions near the anode becomes lower and lower. Therefore, the Mg ions from other places move toward the anode by diffusion or electromigration. According our results, the increased Na2MgY concentration can improve the Mg amount in MAO coatings, indicating that diffusion plays an import role for Mg into coatings.